# Structural and Aggregation Features of a Human κ-Casein Fragment with Antitumor and Cell-Penetrating Properties

**DOI:** 10.3390/molecules24162919

**Published:** 2019-08-12

**Authors:** Olga A. Chinak, Andrey V. Shernyukov, Sergey S. Ovcherenko, Evgeniy A. Sviridov, Victor M. Golyshev, Alexander S. Fomin, Inna A. Pyshnaya, Elena V. Kuligina, Vladimir A. Richter, Elena G. Bagryanskaya

**Affiliations:** 1Institute of Chemical Biology and Fundamental Medicine SB RAS, Novosibirsk 630090, Russia; 2N.N. Vorozhtsov Novosibirsk Institute of Organic Chemistry SB RAS, Novosibirsk 630090, Russia; 3Department of Natural Sciences, Novosibirsk State University, 1 Pirogova Str., Novosibirsk 630090, Russia

**Keywords:** intrinsically disordered protein, dimerization, casein micelle, disulfide bond, β-mercaptoethanol adduct

## Abstract

Intrinsically disordered proteins play a central role in dynamic regulatory and assembly processes in the cell. Recently, a human κ-casein proteolytic fragment called lactaptin (8.6 kDa) was found to induce apoptosis of human breast adenocarcinoma MCF-7 and MDA-MB-231 cells with no cytotoxic activity toward normal cells. Earlier, we had designed some recombinant analogs of lactaptin and compared their biological activity. Among these analogs, RL2 has the highest antitumor activity, but the amino acid residues and secondary structures that are responsible for RL2′s activity remain unclear. To elucidate the structure–activity relations of RL2, we studied the structural and aggregation features of this fairly large intrinsically disordered fragment of human milk κ-casein by a combination of physicochemical methods: NMR, paramagnetic relaxation enhancement (PRE), Electron Paramagnetic Resonance (EPR), circular dichroism, dynamic light scattering, atomic force microscopy, and a cytotoxic activity assay. It was found that in solution, RL2 exists as stand-alone monomeric particles and large aggregates. Whereas the disulfide-bonded homodimer turned out to be more prone to assembly into large aggregates, the monomer predominantly forms single particles. NMR relaxation analysis of spin-labeled RL2 showed that the RL2 N-terminal region, which is essential not only for multimerization of the peptide but also for its proapoptotic action on cancer cells, is more ordered than its C-terminal counterpart and contains a site with a propensity for α-helical secondary structure.

## 1. Introduction

Proteins of human milk, in addition to their nutritional role, have a wide variety of physiological functions, for example, intracellular calcium transport, modulation of the immune system, regulation of cell proliferation, and protection from pathogens [1,2,3]. The milk-derived fusion peptide, ACFP, suppresses the growth of primary human ovarian cancer cells by regulating apoptotic gene expression and signaling pathways [4,5]. Furthermore, it is known that human milk contains proteins and peptides capable of killing cancer cells [6,7,8,9]. Previously, a human κ-casein proteolytic fragment (8.6 kDa) that induces apoptosis in human adenocarcinoma MCF-7 cells with no cytotoxic activity toward normal cells has been found [10] and is named lactaptin.

Among the number of recombinant lactaptin analogs, the most promising is the protein (called RL2, 14 kDa) containing the complete amino acid sequence of lactaptin (Figure 1). When the mechanism of cytotoxic action of RL2 was studied elsewhere, it was shown that this protein is effectively internalized into the cytoplasm of human cancer cells partly by lipid raft–mediated pinocytosis and partly by direct penetration through the plasma membrane [11]. RL2 interacts with cytoskeleton proteins (α- and β-tubulin and α-actinin I) and induces apoptotic death of human cancer cells via mitochondrial and death receptor pathways. The apoptotic action of RL2 is accompanied by a change in the mRNA and protein expression of the apoptosis-related Bcl-2 gene [12,13]. At present, preclinical trials of the biological therapeutic agent “lactaptin” designed on the basis of RL2 [7], have been successfully completed.

Our data on the cell penetration mechanism of RL2 and its primary structure suggest that this peptide can be assigned to the class of cell-penetrating peptides (CPPs). CPPs are known for their effective penetration into cells and their capacity for the intracellular delivery of cargo molecules. The penetration of CPPs into a plasma membrane is assumed to depend on their structural features and disorder-to-order transition after a membrane interaction. Taken together, our results suggest that RL2 is a CPP that has cytotoxic activity toward mammalian cancer cells.

Caseins were some of the first proteins recognized as functional but unfolded [14]. Most of the casein studies have been performed on bovine proteins. In the present work, we study a large proteolytic fragment, which corresponds to 66.5% of human milk κ-casein (amino acid [aa] positions 24–134 [15]) and additionally contains the His-tag GGSHHHHHH at the C terminus and Met as the first amino acid residue. Thus, RL2 should not have rigid secondary structure. A high probability of disorder for its sequence is predicted by the data obtained by the DISOclust method [16] (see Appendix A). Another noteworthy feature of the casein family, which includes secreted, calcium (phosphate)-binding phosphoproteins, is the formation of stable ~200 nm colloidal micelles in water or milk. It is known that all caseins are prone to self-association and in some cases to the formation of amyloid fibrils [14,17]. In this regard, a question arises as to how much RL2 can mimic the aggregation properties of its parent class of proteins.

Intrinsically disordered proteins (IDPs) are involved in a wide range of essential biological processes, especially signaling and regulation, as well as pathological conditions such as cancer, neurodegenerative, and autoimmune diseases. Accordingly, in recent years, much attention has been focused on IDP research [18]. Nonetheless, in view of the lack of ordered structure, there are special methods for the analysis of disordered proteins [19]. NMR spectroscopy is the most powerful method for characterizing local structural preferences and dynamics of IDPs in solution [20,21,22,23,24,25]. In this context, NMR observations allow researchers to obtain information about IDP structural features on a residue-specific basis [26,27]. In all previous work, NMR research has been performed on model proteins and bovine proteins. In this paper, we present the first NMR application to study a large fragment of human κ-casein protein.

The aim of this work was to study the structure of the recombinant protein RL2 in the same form as in the above-mentioned therapeutic agent. We characterized the structural and aggregation properties of recombinant protein RL2 in solution by a combination of low-resolution methods: circular dichroism (CD), dynamic light scattering (DLS), and atomic force microscopy (AFM) with a high-resolution method (NMR spectroscopy), in order to understand whether and how these properties influence RL2 activity and its cell penetration efficiency. NMR measurements performed in this study revealed that RL2 is a mixture of a disulfide-bonded homodimer and a monomer covalently bound to β-mercaptoethanol (BME) via the S–S bond. The analysis of RL2 preparation showed that BME adduct with RL2 is formed in the process of preparation of the recombinant protein. As the existence of BME adduct was not known before it was interesting to understand whether BME adduct influence on biological activity and aggregation of RL2.

## 2. Experimental Section

### 2.1. Expression and Purification of [U-^13^C,^15^N]-RL2 or [U-^15^N]-RL2

RL2 labeled with ^13^C and ^15^N isotopes were expressed in the *E.coli* BL21(DE3) strain (Stratagene). The culture was grown in Silantes rich growth media (for *E.coli)* (^13^C, 98%; ^15^N, 98%) supplemented with 3 mM Na_2_HPO_4_, 2.5 mM KH_2_PO_4_, 1 mM NaCl, and ampicillin 150 mg/L. Upon reaching an optical density at 600 nm, 0.4, isopropyl β-D-1-thiogalactopyranoside (IPTG) was added into the culture to induce recombinant protein expression under the same conditions. Cells were pelleted from culture medium and re-suspended in lysis buffer. RL2 was purified by affinity chromatography using Imac sepharose, ion-exchange chromatography on an DEAE Sephadex A-25 and SP Sepharose. All buffers contained 0.1 M Tris-НСl pH 7.5, 2 M urea, 10 mM β-mercaptoethanol. Then RL2 was dialyzed against water.

### 2.2. Labeling of [U-^15^N]-RL2 with a Paramagnetic MTSL Probe

A total of 1.5 mg of RL2 was dissolved in a buffer (4 M guanidinium chloride, 20 mM EDTA, 0.1 M Tris-HCl pH 6.0). A 1000-fold excess of tris(2-carboxyethyl)phosphine TCEP was added. After 3 h incubation at 55 °C, TCEP was removed using an Amicon Ultra 15 concentrator with a 3 kDa cutoff (Millipore, Burlington, MA, USA). Then, a 135-fold excess of *S*-(1-oxyl-2,2,5,5-tetramethyl-2,5-dihydro-1H-pyrrol-3-yl)methyl methanesulfonothioate (MTSL) was added to the protein and incubated overnight at 50 °C. Next, the unreacted probe and buffer components were removed in a concentrator with a 3 kDa cutoff.

### 2.3. Chromatographic Separation of the RL2 Dimers

The dimers were separated from RL2 by gel-filtration on Superdex 200 (GE Healthcare, Stockgolm, Sweden) using a solution of 0.15 M NaCl, 50 mM NaAc (pH 5.5) with no reducing agents. Then, the buffer was removed in a concentrator with a 3 kDa cutoff. The molecular weight of the proteins obtained was estimated by its mobility by non-reducing SDS-PAGE. (Appendix A)

### 2.4. EPR Measurements

X-band EPR spectra were recorded using a commercial Bruker EMX EPR spectrometer (9 GHz, Bruker Spectrospin, Karlsrue, Germany). The spin-labeled RL2-MTSL (5 × 10^−4^ M) was dissolved in a buffer consisting of 50% D_2_O and 50% deuteroglycerol acetic acid at pH 3.9, and at pH 7.5, 150 mM NaCl was added, and the solution was placed in a capillary of a 0.5 mm diameter and analyzed at 300 K or shock-frozen at 132 K. The external magnetic field was modulated with a frequency of 100 kHz and an amplitude of 1 G. The microwave power was adjusted to 2.0 mW and all measurements were conducted with a time constant of 20.48 ms. Simulations of the EPR spectra were carried out by means of the software package Easy Spin (www.easyspin.org).

### 2.5. NMR Sample Preparation, Data Collection and Processing

RL2 samples were concentrated in a concentrator with a 3 kDa cutoff and diluted in a H_2_O–D_2_O (90:10) mixture at pH ≈ 3.5 (HCl-adjusted for reproducible ^1^H-^15^N Heteronuclear Single Quantum Coherence (HSQC) results) or in 20 mM acetate buffer (pH 3.9). To all the samples, 50–100 µM monosodium salt of 4,4-dimethyl-4-silapentane-1-sulfonic acid (DSS) was added as a reference standard for direct and indirect shift calibration. For NMR measurements, RL2-MTSL was dissolved in 20 mM sodium acetate buffer (pH 3.9).

All the spectra were acquired at 20.1 °C on a Bruker Avance 600 MHz spectrometer (Bruker Spectrospin, Karlsrue, Germany) equipped with a 5 mm {^15^N,^13^C}^1^H triple-resonance z-gradient probehead. The double- and triple-resonance experiments were conducted for the sequence-specific backbone and partial side chain assignments and included 2D ^15^N-HSQC, 3D CBCA(CO)NH, CBCANH, HNCO, HN(CA)CO, HNCA, HN(CO)CA, ^15^N-TOCSY, and ^15^N-NOESY. For ^15^N T_1_, spectra with relaxation delays of 50, 110, 180, 256, 346, 460, 600, 1200, and 1800 ms were acquired, and for ^15^N T_2_, the spectra were recorded by means of relaxation delays of 0, 0, 16, 33, 49, 66, 98, and 147 ms. Vicinal couplings between HN(i) and HA(i) spins were measured by 3D HNHA. The spectra were processed on Bruker Topspin and analyzed in CCPNMR2 [28].

### 2.6. DLS Analysis

The Z-average diameter of RL2 particles was determined by dynamic laser light scattering using a Zetasizer Nano ZS (Malvern Instruments Ltd., Malvern, UK) at a wavelength of 623 nm and 25 °C. All the buffers were centrifuged and filtered (0.22 μm pore size). Measurements were performed in a 40 μL micro cuvette ZEN0040. Samples were diluted to RL2 concentration of 0.28 mg/mL in 25 mM MES buffer (pH 5.5–6.5) or 25 mM Tris-HCl buffer (pH 7.0–8.0) before determination. Mean particle diameters were calculated as the average of at least triplicate measurements. The presented sizes are weighted by intensity.

### 2.7. AFM

Ten microliters of an RL2 suspension was deposited onto a freshly made mica slide (1 × 1 cm) in 25 mM MES buffer pH 5.5. The solution was incubated for 30 s, and then the slide was washed with deionized water and dried for 1 min under a gentle argon stream. AFM experiments were carried out on a Multi-Mode 8 (Bruker, Karlsrue, Germany) with a scanning area of 10 × 10 and 1.8 × 1.8 µm. Images were captured in semicontact mode under atmospheric conditions using a diamond-like carbon NSG series AFM cantilever (NT-MDT, Zelenograd, Russia) with the radius of the tip curvature 1–3 nm. The AFM images were processed in NanoScope Analysis (Bruker, Karlsrue, Germany).

### 2.8. CD Spectroscopy

This analysis of RL2 was performed by means of a Jasco 600 spectropolarimeter (Jasco Instruments, Tokyo, Japan) in a 0.1 cm light path quartz cuvette (Starna, Essex, UK). Data were obtained at wavelengths 250–185 nm at 1 nm intervals in three individual scans. RL2 samples (0.25 mg/mL) were dissolved in 150 mM NaCl in the presence or absence of 50% trifluoroethanol (TFE).

## 3. Results and Discussion

### 3.1. Study of RL2 Cytotoxic Activity

It has been demonstrated previously by nonreducing SDS-PAGE [29] that recombinant protein RL2 consists of a monomeric peptide (14 kDa) and a covalent disulfide-bonded homodimer (28 kDa), the latter is formed by the disulfide S–S bond between Cys8 residues. These forms exist in the practical immutability ratio with time in solution. But the causes of this equilibrium remained unclear. Proteins with active cysteine are known to produce adducts with various low-molecular-weight compounds [30,31]. In our case, the only option was an adduct with β-mercaptoethanol (BME) [32,33], which is formed in the process of preparation of the recombinant protein. Indeed, we have been able to prove this idea clearly by NMR (see below) and to obtain RL2 only in the monomeric form with free cysteine when using TCEP as a strong reducing agent [32]. Thus, we show that a recombinant analog of lactaptin, RL2, represents a mixture of a disulfide-bonded homodimer and a monomer covalently bound to β-mercaptoethanol via the S–S bond (RL2-BME). We suppose that the RL2-BME adduct is stable because of the dimer to RL2-BME ratio is constant with time in solution.

To analyze the relation between RL2 forms and their activity, we separated dimeric RL2 by chromatography and measured the cytotoxic activity of RL2 and dimeric RL2 against human lung cancer cells A549 by an MTT assay as described previously [34] (Figure 2).

As shown in Figure 2a, there was no significant difference in the viability of A549 cells treated with either RL2 or dimeric RL2; their half-maximal inhibitory concentration values (IC50) were 0.26 and 0.25 mg/mL, respectively. Thus, adduct RL2-BME and dimeric RL2 have equal cytotoxic activity toward cancer cells.

Because RL2 dimerization did not increase its activity, we obtained new recombinant protein RL2^Ser^ with Cys substitution by Ser and measured the cytotoxic activity of this new recombinant protein against MDA-MB-231 cells by the MTT assay (Figure 2b). The data showed that the IC50 of RL2 was 0.39 mg/mL, and RL2^Ser^ did not reach IC50 within the range 0–0.5 mg/mL. Thus, the Cys-to-Ser substitution inhibits the RL2 cytotoxic activity. A possible explanation for the loss of cytotoxic activity for RL2^Ser^ could be that Cys thiol exchange plays a role in the internalisation mechanism [35,36]. However, in the case of RL2, Cys is located in the region responsible for RL2 multimerization. Since in cell medium RL2 is highly aggregated, we suggest that the majority of Cys residues are exposed inside the RL2 particles. It is more likely that RL2 interacts with the negative charged cell surface because of its positive charge.

Thus, Cys is an essential amino acid residue for RL2 activity, and RL2-BME has the same cytotoxic activity toward cancer cells as dimeric RL2 does. Therefore, we hypothesized that both RL2-BME and the RL2 dimer could be reduced by intracellular conditions, with a release of monomeric RL2. To estimate the contribution of BME to RL2-BME activity, we measured its cytotoxic activity. Even after treatment at a high concentration (1 mM), cell viability was 86.9 ± 4.9% (mean ± SD). Thus, BME does not contribute to the cytotoxic activity of RL2.

Taking the above results into account, we assumed that attachment of the small organic molecule to the cysteine via the S-S bond should not decrease RL2 activity.

### 3.2. RL2 Aggregation Properties. DLS and AFM Analyses at Different pH Levels

It is known [14], that κ-casein stabilizes micelle formation, preventing casein precipitation in milk. All caseins consist of “sticky” P,Q-enriched sequences, which are one of the reasons for the ability of their molecules to easily associate with each other [14]. Such sequences with high P+Q content stimulate protein–protein nonspecific interactions, which are sometimes referred to as “promiscuous” [37]. Because of their easy association with each other, casein protein molecules can form amyloid fibrils or nearly spherical particles [14,18,38]. We performed DLS analysis of the particle diameter distribution of RL2 under different conditions.

The size and shape of RL2 particles was estimated here by DLS and AFM. The results are summarized in Table 1 and presented in SI. RL2 size distribution in the absence of NaCl was found to have a high polydispersity index (PDI = 0.43 ± 0.80), which means that the diameter of particles is not homogeneous. The diameter distribution of RL2 particles at pH 5.5 and 6.0 has a predominant peak at 7 nm and a minor peak at 215 nm. At pH ≥ 6.5, the proportion of RL2 particles with a larger diameter became predominant.

In the presence of NaCl at pH 5.5 and 6.0, the diameter distributions of RL2 particles have peaks with mean values at 7 and 220 nm. At pH ≥ 6.5 and physiological ionic strength, low polydispersity (PDI = 0.22 ± 0.13) was observed, while the peaks at 7 nm were not detectable. The change in pH from 5.5 to 8.0 led to a dramatic increase in the diameter (a shift of the second peak from 196 to 721 nm). In addition, it should be noted that in all the above cases, we observed insignificant peaks corresponding to diameters >1 μm.

We found that the oligomer ratio depends on pH. Increasing pH led to RL2 oligomerization, which was probably caused by a decrease in the positive charge and repulsion between RL2 particles; therefore, noncovalent interactions of P,Q-rich RL2 sequences became possible. The presence of physiological ionic strength caused RL2 to oligomerize at lower pH as compared with the absence of NaCl, suggesting that ionic strength and pH are the key factors for RL2 polymerization. We observed that under the conditions close to the extracellular environment, RL2 consists of large 700 nm oligomers, whereas at pH 5.5 (corresponding to early endosomes), RL2 is predominantly in monomeric/dimeric forms, and its oligomers have a size of ~200 nm.

AFM revealed the presence of globular or ovoid small particles and some larger particles (Figure 3), which correspond to monomeric/dimeric RL2 and oligomeric RL2, respectively. It was found that the RL2 structures have single-object widths of 5 ± 2 nm (not tip-corrected) and height maxima of 1.6 ± 0.3 nm, whereas the RL2 oligomer structures are characterized by object widths in the range 10–25 nm and height maxima in the range 3–8 nm. In spite of the small number of larger particles, they may represent a significant amount of RL2 by weight. RL2 can form micelles as well as κ-casein can, in contrast to κ-casein, which is known for its propensity for fibrillation [14,39,40]. RL2 fibrils were not observed in our experiments.

### 3.3. EPR Analysis at Different pH Levels—Formation of Aggregates at Physiological pH

We studied the tendency for monomer aggregation by means of EPR and MTSL-labeled RL2. The concentration of MTSL spin labels at the cysteine position of RL2 according to EPR spectroscopy was 4 × 10^−4^ M, and the concentration of protein as determined using the Bradford assay was 5 × 10^−4^ M. Thus, effectiveness of spin labeling was 0.8. Spin-labeled RL2 was investigated using continuous-wave (CW) EPR spectroscopy at room temperature (300 K) and low temperature (132 K), and at different pH levels (Figure 4). At low pH, 3.9, the CW EPR spectrum showed a 96% fraction of MTSL nitroxide attached to RL2 characterized by low mobility with rotation correlation time 1.95 ns as well as 4% of free MTSL nitroxide with high mobility with rotation correlation time 0.27 ns.

The magnetic resonance parameters were obtained from simulations of EPR spectra of frozen solution (Figure 4C) and were used in simulation of EPR spectra at room temperatures (Figure 4A,B). The alteration of pH from low, 3.9, to pH 7.5 leads to significant change of EPR spectrum of the main fraction and appearance of very wide line with very short electron spin relaxation time. At the same time, the intensity and shape of the EPR spectrum of free nitroxide remains the same.

As presented in Figure 4B, at 300 K, the signals of the main fraction—RL2-MTSL for the sample prepared under physiological pH conditions—are greatly broadened compared to the signals of the main fraction under acidic pH conditions (Figure 4A,C). At the same time, the signals of the free label stay virtually unchanged. At low temperature, 132 K, signals of the main fraction are not observed at pH 7.5. These results are in good agreement with the fact that under physiological pH conditions, most of the protein forms aggregates, which greatly broaden the EPR spectra because of two factors. On the one hand, the large size of aggregates corresponds to slow rotation and long correlation times. On the other hand, modulation of the dipole–dipole interaction and the exchange interaction between spin labels in aggregates lead to a short electron spin relaxation. We failed to observe Pulse Electron Double Resonance (PELDOR) of the samples in the acidic condition; this result is explained by a substantially lower amount of non-covalent oligomers in the solution under these conditions. For the sample at pH 7.5, it was impossible to obtain the PELDOR spectrum owing to short T_2_ relaxation time; this result can also be explained by the presence of an exchange interaction between spins and the wide distribution of the dipole–dipole interaction in aggregates. Thus, we can conclude that a significant proportion of the protein at physiological pH forms large aggregates.

### 3.4. CD Analysis—The Secondary Structure of RL2 and Its Thermodynamic Stability

To estimate RL2 secondary structure and its thermodynamic stability under different conditions, CD spectra in the far-UV region were recorded in buffered saline and in 50% TFE at temperatures of 10–90 °C. CD spectra of RL2 are depicted in Figure 5 and are characterized by a minimum at 198 nm, which indicates a fully or partially disordered peptide conformation and suggests that RL2 is an IDP. Though RL2 is proline-rich protein, it does not have polyproline helix structure PPII, because of the absence of the positive peak at 218 nm in CD spectrum. Changing the solution temperature from 10 °C to 90 °C led to a decrease in negative intensity of the 198 nm peak and to the emergence of 206 and 222 nm peaks, indicating that the structure of RL2 becomes more ordered: the α-helix content increases. Upon cooling of the denatured RL2, its original structure got restored, and there were no significant differences in RL2 structure before and after temperature denaturation. Such folding upon heating looks counterintuitive for structured proteins, but it is common among IDPs [41]. Thus, temperature denaturation–renaturation of this peptide is a reversible process.

CD spectroscopy performed in the presence of hydrophobic TFE suggested that TFE promotes significant conformational changes in the secondary structure of RL2, thereby leading to peptide folding. Two negative peaks, at 208 and 222 nm, point to an increase in α-helix content. In 50% TFE, the secondary structure of RL2 thermodynamically stabilized, and the CD spectra manifested the highest correlation with the secondary structure in the aqueous solution at 90 °C. To estimate the level of residual RL2 secondary structure, we calculated the [θ]_200_/[θ]_222_ ratio for CD spectra [42]. We found that in buffered saline, RL2 has a coil-like structure, whereas in hydrophobic TFE, it has a premolten globule state. Thus, CD spectroscopy indicates that RL2 has an intrinsically disordered structure capable of partial folding in hydrophobic environment. Such partial folding on the plasma membrane is described for disordered CPPs, and this property is believed to allow these peptides to penetrate lipid cell membranes.

It is commonly known that κ-casein, from which RL2 is derived, carries Ca^2+^ across the wall of the distal small intestine. Moreover, our data (in press) suggest that RL2 also mediates the delivery of plasmid DNA and siRNA into human cells. Thus, RL2 is capable of cross-membrane transport just as κ-casein is, probably due to such folding.

### 3.5. NMR Analysis. Dynamics and PRE Data on RL2 and MTSL-RL2

Because of the presence of BME-RL2 and of the covalent disulfide-bonded homodimer of RL2, native protein represents a complex system in solution and is difficult to study by NMR. Furthermore, as shown above by DLS, AFM, and EPR data, large casein-like micelles can form, which contain protein amounts comparable to those in solution. An additional complication is the fact that RL2 is a proline-rich protein and contains 20 proline residues (17%) in its sequence. This property makes it very likely to contain minor conformers in solution (cf [43]), taking into account its IDP properties. In an NMR study of the peptide corresponding to the fragment (aa 108–125) of the human κ-casein full sequence [15] (for RL2 86–103) and containing four prolines, additional peaks were observed attributed to a minor isomer [44]. An NMR study on casein proteins containing casein micelles has revealed the presence of sufficiently narrow lines in their spectra [45]. This finding can be explained by their IDP nature and increased times of T_2_ relaxation as compared with the ordered proteins as well as by the presence of a sufficient number of small oligomers or the monomeric form of these proteins in solution. Indeed, sufficiently narrow lines are observed in the NMR spectra of RL2.

At physiological pH, most of the amide group signals of RL2 were absent in the *^1^H-^15^N HSQC* spectrum owing to the exchange with water (see Appendix A). When the solution was acidified, the rate of exchange with the solvent decreased, and the missing cross-peaks appeared in the spectrum, while the dispersion of amide signals on the proton axis was also improving. Good-quality spectra of RL2 suitable for assignment were obtained at pH values close to 4 and temperature 20.1 °C. The effect of pH on structural properties of IDPs is small until pH reaches levels close to their pI values [19]. The experimental pI value of RL2 is 8.0; therefore, one should expect that its structural properties at acidic pH will not differ much from properties under physiological conditions.

All signals of amide groups are visible in the ^1^H-^15^N HSQC spectrum (Figure 6A), and nitrogen shifts have good dispersion; these circumstances allowed for complete assignment of backbone atoms. When reducing agent TCEP was added to the sample, the main differences in the spectra were only in the positions of the cross-peaks of the residues adjacent to cysteine (Figure 6B). Moreover, we noted the emergence of BME signals in the ^1^H NMR spectrum (see Appendix A) and a change in chemical shifts CA (from 55.40 to 58.35 ppm) and CB (from 41.10 to 28.00 ppm) of Cys8 to the values corresponding to its -SH form.

To simplify the studied system and compare it with the original one, an MTSL-labeled sample of RL2 (at cysteine 8) was also obtained. In this case, the formation of a disulfide-bonded homodimer is impossible, which allows us to exclude its influence on the system and enables obtaining additional information about long-range structural features by PRE analysis [46]. Surprisingly, the ^1^H-^15^N HSQC spectrum of RL2 is almost identical to that of diamagnetic MTSL-RL2 (Figure 6). The same is true for the ^15^N-relaxation data of both (Figure 7A–C). Eventually, ^15^N backbone relaxation of the Cys8-labeled protein coincides with that of the native one (Figure 7A–C). Some minor differences in the dynamics of the His-tag and a region near the Asp12 site can be explained by slightly different conditions. In fact, the chemical shifts in these regions are also most perturbed and secondary structure propensities (SSPs) [47] are slightly different (Figure 8A).

Average T_1_ = 0.7 s in both cases is typical for a protein of 14 kDa at this temperature and field strength (see Appendix A). Taking into account the presence of oligomeric RL2 even under acidic conditions (Table 1) and the absence of non-covalent RL2-MTSL oligomers under the same conditions it is reasonable to assume that (i) either NMR spectra of monomers and dimers are similar or (ii) the signals from the dimers in NMR spectra are absent because of its greater tendency to aggregate compared to monomer even under acidic conditions (confirmed by DLS). As the relaxation times of monomers and dimers are different due to different sizes, it should be seen in NMR spectra which is not the case. Thus, from our point of view the second proposal is much more reasonable. This means that the monomeric form in RL2 is the BME-adduct and only this form is observed in NMR spectra, while the disulfide-bonded homodimer should be almost exclusively located in large oligomers, which are hidden from NMR spectroscopy. This situation provides a unique opportunity to study the monomeric BME-RL2 adduct by NMR directly in the mixture under nonreducing conditions. Probably because of the greater propensity to oligomerize, the dimer is mainly located in large oligomers that are hidden from NMR analysis, while the BME adduct of RL2 is protected from this process. At the same time, the presence or absence of BME at Cys8 has virtually no effect on RL2 residual structure (Figure 8B). Nevertheless, more peaks were observed in the spectrum in comparison with the single protein. These mostly less intense signals were also observed in the spectra of MTSL-labeled RL2, which means that they are related to additional conformers of the protein that are due to *cis*-prolines rather than disulfide-bonded homodimer contamination. All the assigned regions belong to the main conformation of RL2 because all the prolines (with the exception of 82 and 99, which are at Pro-Pro sites and are hidden for standard NH-detecting 3D experiments) have chemical shifts of CB atoms in the range of 31.5–32.2 ppm and therefore are in the trans-conformation [48].

Nuclear spin relaxation rates are used for characterization of the dynamics of biomacromolecules because of their sensitivity to molecular motions [24,49]. HetNOE data suggest the presence of a more ordered site at the N terminus of RL2 (Figure 7C). N-terminal region 1–43, which corresponds to the aa 23–66 fragment of human k-casein, is essential not only for multimerization of the peptide but also for the apoptotic action of lactaptin on MCF-7 cells [29]. SSP scores point to the presence of a region with an α-helix propensity: ^30^Asn-Ser-Tyr-Pro-Tyr-Tyr-Gly-Thr-Asn-Leu-Tyr^41^. Thus, this site may be responsible for the interaction of RL2 with the cell surface or its possible intracellular targets: proteins of the cytoskeleton.

## 4. Conclusions

In summary, we showed that a recombinant analog of lactaptin, RL2, represents a mixture of a disulfide-bonded homodimer and a monomer covalently bound to β-mercaptoethanol (BME) via the S–S bond. These forms of RL2 in solution manifested different behaviors. Whereas the disulfide-bonded homodimer is more prone to assemble into large aggregates, the monomer predominantly forms single particles, which were observed in NMR spectra of RL2.

On the basis of NMR analysis of RL2 samples it is shown that that the *N*-terminal region of RL2, which is essential not only for multimerization of the peptide but also for the apoptotic action of lactaptin on cancer cells, is more ordered than its C-terminal counterpart and contains a site with a propensity for α-helical secondary structure. Moreover, Cys8 in this sequence is an essential residue for RL2 activity.

Using the combination of CD, DLS, AFM and NMR spectroscopy, it was found that RL2 is an intrinsically disordered peptide able to adopt a thermodynamically stabilized partially folded structure in a membrane-mimetic environment. These structural variations allow RL2 to penetrate through the cell lipid membranes. RL2 forms an insignificant amount of aggregates and can form micelles as well as κ-casein, but unlike the latter, RL2 is unable to form fibrils.

The N-terminal region (aa 1–43), responsible for RL2 multimerization and partially for its activity, includes the only region with an α-helical propensity (aa 30–41). Moreover, Cys8 in this sequence is an essential residue for RL2 activity. NMR analysis of RL2 samples confirmed the presence of RL2-BME adducts. Our results may form the foundation for future research into the mechanism of RL2 anticancer activity by NMR spectroscopy.

The data obtained reveal that RL2 aggregation properties, which are important for intravenous injection, may be changed by the introduction of a small organic molecule at the cysteine position.

## Figures and Tables

**Figure 1 molecules-24-02919-f001:**
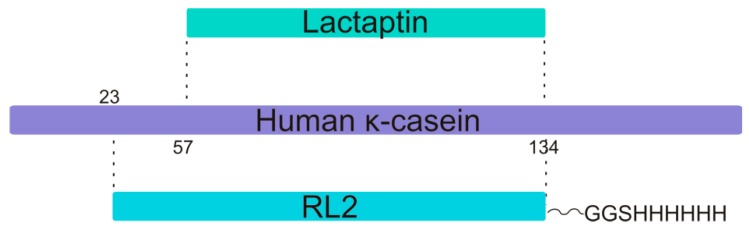
The relation of the primary structures of human κ-casein, its proteolytic fragment lactaptin, and a recombinant analog RL2. Residue Cys8 in RL2 recombinant fragment corresponds to position Cys30 of human kappa-casein polypeptide (P07498 entry).

**Figure 2 molecules-24-02919-f002:**
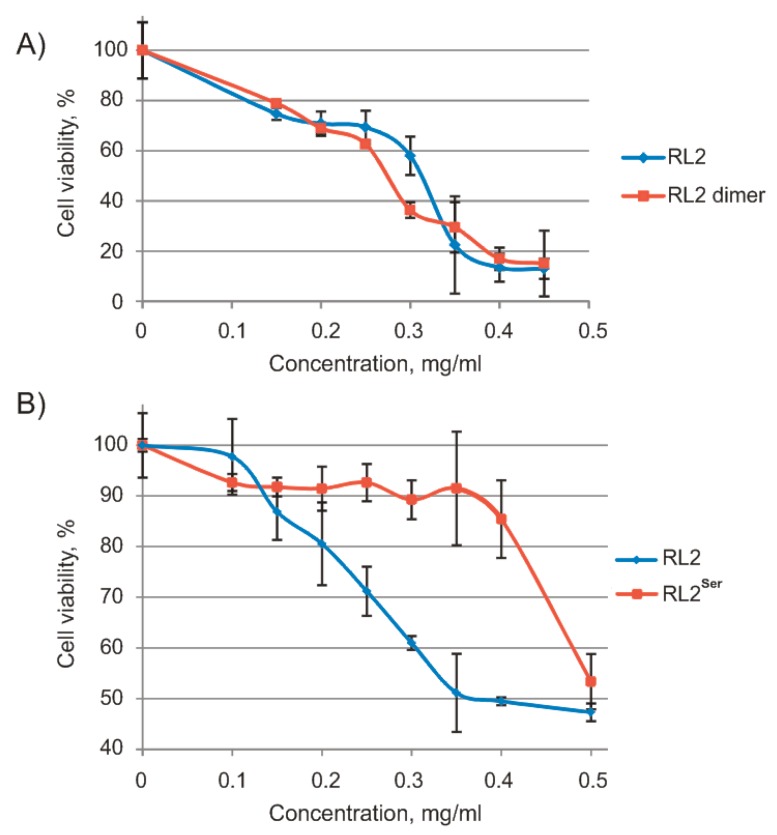
Viability of human cancer cells treated with RL2, dimeric RL2, and RL2^Ser^ for 48 h (MTT assay). (**A**) A549 cells treated with different RL2 or RL2 dimer concentrations; (**B**) MDA-MB-231 cells treated with different concentrations of RL2 and RL2^Ser^. Differences between the means control and experimental groups of MDA-MB-231 cell viability are statistically significant in *p* < 0.02.

**Figure 3 molecules-24-02919-f003:**
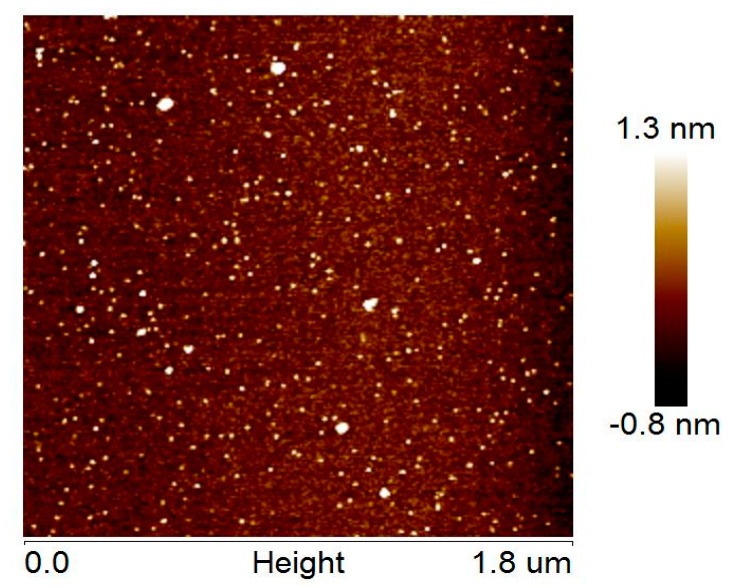
An AFM image of RL2 particles.

**Figure 4 molecules-24-02919-f004:**
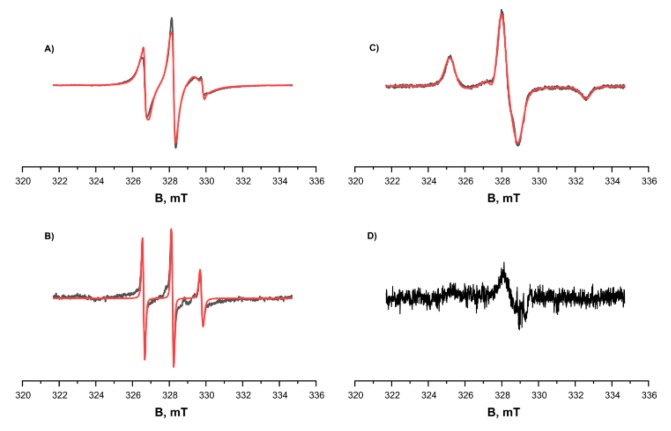
CW EPR spectra of RL2-MTSL at different temperatures and different pH levels (black curve: experimental, red one: simulated). (**A**) and (**C**) pH = 3.9, acetate buffer, 50% D_2_O, 50% deuteroglycerol. (**B**) and (**D**) pH = 7.5. (**A**) and (**B**) T = 300 K. (**C**) and (**D**) T = 132 K. Parameters of the simulation were as follows: g_xx_ = g_yy_ = 2.0071, g_zz_ = 2.0022, A_xx_ = A_yy_ = 0.5 mT, A_zz_ = 3.7 mT, τ_corr_ = 1.95 ns. The additional curve in spectrum **A** corresponds to free nitroxide at a concentration ~4% of the attached MTSL with correlation time τ_corr_ = 0.27 ns and is clearly visible in spectrum (**B**).

**Figure 5 molecules-24-02919-f005:**
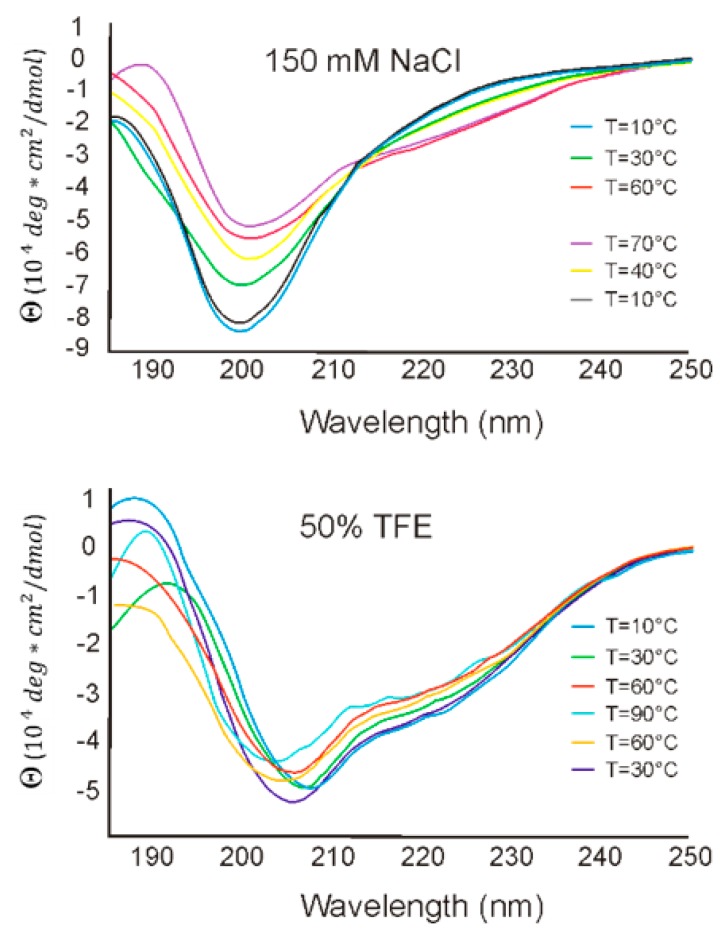
CD spectra of RL2 in aqueous and hydrophobic (trifluoroethanol; TFE) solutions.

**Figure 6 molecules-24-02919-f006:**
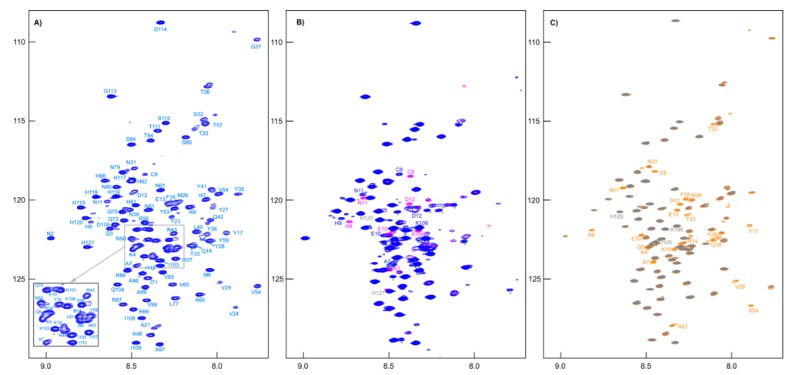
(**A**) The *^1^H-^15^N HSQC* spectrum of [U-^13^C,^15^N]-RL2 (~1 mM, pH 3.5). (**B**) Overlapping *^1^H-^15^N HSQC* spectra of [U-^13^C,^15^N]-RL2 (~1 mM, acetate buffer, pH 3.9) before (blue peaks) and after (pink peaks) addition of TCEP (an asterisk indicates peaks of impurities). (**C**) Overlapping *^1^H-^15^N HSQC* spectra of MTSL-[U-^15^N]-RL2 (~0.1 mM, acetate buffer, pH 3.9) before (gray peaks) and after addition (orange peaks) of ascorbic acid.

**Figure 7 molecules-24-02919-f007:**
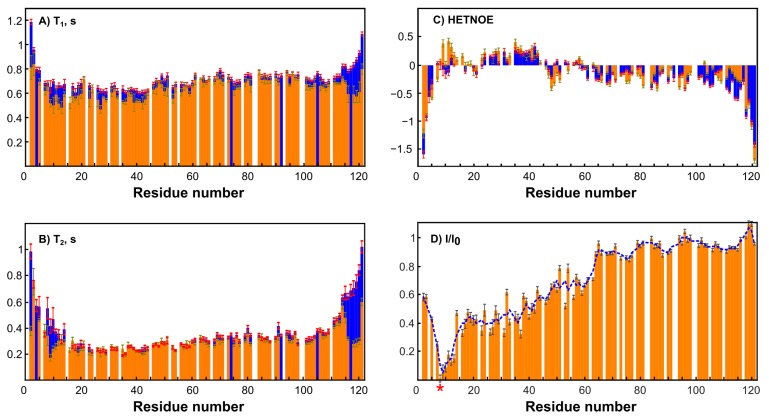
Dynamics and PRE data on RL2 and MTSL-RL2. Error bars represent random noise error of fitting the experimental data. (**A**) A comparison of T_1_ values of RL2 at pH 3.5 (blue) and diamagnetic MTSL-RL2 (orange). (**B**) The comparison of T_2_-values of RL2 at pH 3.5 (blue) and diamagnetic MTSL-RL2 (orange). (**C**) A comparison of hetero NOE values of RL2 at pH 3.5 (blue) and diamagnetic MTSL-RL2 (orange). (**D**) Integral ratios of peaks in the HSQC paramagnetic and diamagnetic spectra of MTSL-RL2. The blue dashed curve is a three-point moving window average for the best trend view. The asterisk indicates the MTSL-label site (Cys8).

**Figure 8 molecules-24-02919-f008:**
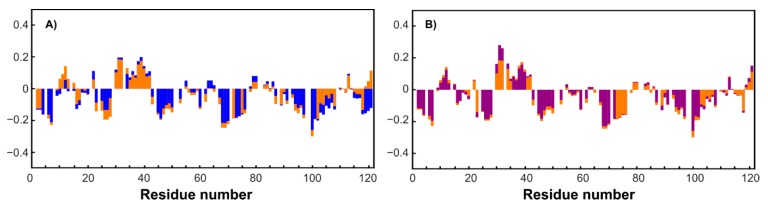
A comparison of SSP scores. (**A**) Blue bars denote RL2 at pH 3.5, and orange ones correspond to acetate buffer, pH 3.9. (**B**) Orange and purple bars denote RL2 and RL2 with TCEP added, respectively, at pH 3.9 in acetate buffer.

**Table 1 molecules-24-02919-t001:** The dependence of RL2 particle diameters (in nm) on pH and on the presence of physiological ionic strength.

		pH 3.9	pH 5.5	pH 6	pH 6.5	pH 7	pH 7.5	pH 8
No NaCl	Peak 1	11.7 ± 2.1	7.2 ± 0.7	6.9 ± 0.5	-	7.0 ± 0.9	7.0 ± 0.8	-
	Peak 2	288.4 ± 129.3	220.3 ± 60.8	210.5 ± 55.6	254.9 ± 58.6	147.8 ± 27.3	161.5 ± 39.4	170.7 ± 21.1
With NaCl	Peak 1	10.5 ± 1.2	6.6 ± 0.1	6.2 ± 0.3	-	-	-	-
	Peak 2	428.6 ± 167.8	196.1 ± 25.8	241.7 ± 40.8	255.9 ± 52.7	295.2 ± 18.1	699.4 ± 59.2	721.1 ± 35.0

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
