# Peer review of "Structural and Aggregation Features of a Human κ-Casein Fragment with Antitumor and Cell-Penetrating Properties"

_molecules, 2019, doi:10.3390/molecules24162919_

Round 1
Reviewer 1 Report
The paper: ” Structural and aggregation features of a human k-casein fragment with antitumor and cell-penetrating properties” is well-written, interesting and investigates the IDP mentioned using several different techniques. The reason for the study as well as the experiments are properly described. In the results and discussion and conclusions, the reasoning can be easily be followed and the final conclusions are explained by the experimental data.
Some small corrections to the text in the introduction:
Line 14: replace “have” with “had”.
Line 46: replace “are” with “have been”.
Line 54: add “the” before casein.
Line 72: add “the” before analysis.
I recommend that the paper can be published as is with the small errors corrected.
Author Response
We are very thankful to reviewer for comments and we took into account and corrected all of them.
Reviewer 2 Report
The paper by Olga A. Chinak et al. describes characterization of recombinant peptide RL2 which is a fragment of kappa-casein. Authors used many powerful methods, but the aim and logics of the work is unclear for me. Authors obtained some heterogeneous mixture and analyze their components and the mixture “as is”, but the reason of these experiments is unclear. It is also unclear which form is analyzed – the mixture, monomers or dimers. Judging from two independent graphs in Fig.2A and NMR data, it is possible to obtain the individual forms (monomers bound with BME and dimers), but if so, why the Authors needed to study the mixture by many complicated methods? What new significant information was obtained? I cannot understand the aim of this study.
Authors claim medicinal use of their peptide. May be, it would be more interesting to change production conditions to obtain homogeneous fraction of RL2 instead of complicated study of the details of the mixture behavior?
Second, some conclusions are poorly corroborated with the data:
Interpretation of CD data is questionable. It looks like heating in 150mM NaCl solution caused appearance of beta-structures of Pro-helices but not alpha-helices: no peaks at 208 and 222 nm can be observed.
Authors conclude that Cys8 is important for activity. If so, why formation of disulfide bond does not influence on the activity (Fig. 2, A)?
Is the RL2-BME adduct stable? If not, intravenous injection is questionable.
Minor:
l. 13-14 – the reference is required.
l. 32-37 – I am afraid such conclusion requires more examples with different proteins, not only the Authors own papers and one more protein.
l.160-166 – this part is unclear. What does the following statement mean? “monomer covalently bound to β-mercaptoethanol via the S–S bond”. Why effect of BME is noteworthy?
ссылки 7-8 – проверить, что там, есть ли новизна, насколько вообще норм сделано.
Fig.2 – to judge about IC50, higher concentrations should be tested to obtain viability closer to 0.
Fig2 A-B – why different cell lines were used?
l.398-399 – the statement about insignificant amount of aggregates contradicts with DLS data. The fact that “RL2 is unable to form fibrils” was not corroborated with experiments.
l.408-409 “RL2 aggregation properties … may be changed by introduction of a small organic molecule at the cysteine position” – it seems to be a common situation for BME in case of intermolecular SS bonds.
Author Response
We are very thankful for reviewer for very thoughtful and valuable comments. We took all of them into account.
Please see the attachement.

Reviewer 3 Report
In this manuscript, O. A. Chinak et al. have carried out biophysical studies to characterize the structural and aggregation properties of the RL2 fragment derived from human kappa-casein. This long fragment is endowed with interesting cell-penetrating and proapoptotic properties on cancer cells, and analogs are currently under investigation in preclinical trials.
The authors have used a comprehensive set of biophysical techniques: DLS, AFM, EPR, CD, and NMR spectroscopy to characterize the conformation and aggregation properties at different scales. They found interesting results regarding the aggregation, dynamics and residual helical folding in the N-terminal part of LR2 protein.
However, in my opinion, the manuscript is flawed with many overstatements and insufficient biochemical characterization. The authors should address the following major concerns.
(1) The protocol of protein preparation and the characterization of the Cys redox status of the LR2 protein suffer from a lack of experimental details.
In section 2.1, a more detailed purification protocol should be provided or additional references to published work should be included.
- For instance, there is no mention of added beta-mercaptoethanol in the protocol, whereas it seems to be a critical parameter for protein preparation.
- In their paper published in Protein J. 2010 20:174, the authors have used a chaotropic agent (guanidinium chloride) for RL2 purification. Is it also the case in this study?
- The conditions for RL2 dimer purification are not clear: have oxidative conditions been used?
- Gel filtration chromatograms showing the separation of RL2 dimer from the monomer should be added in the Supporting Information.
(2) The use of beta-mercaptoethanol yields to complex chemical equilibria that hamper the analysis of results. It would have been preferable to use DTT or TCEP as a reducing agent to avoid the formation of adducts with BME. The authors might also consider alkylating the free Cys with iodoacetamide.
In my opinion, the NMR work performed on RL2-MTSL adduct provides more satisfactory analysis since the Cys state of the protein is much better controlled for this sample.
(3) In the NMR study page 10, the authors state that “intense signals from the dimers in the spectra [are] absent”. This statement is not clear. In which protein preparation do the authors expect to have a significant dimer population?
I recommend to record an 1H-15N HSQC spectrum of the purified dimeric form to reveal any differences with the monomeric form.
Furthermore, it is written in conclusion, page 12 (line 405), that “NMR analysis of RL2 samples confirmed the presence of S-S homodimers”. I could not find any data in the manuscript to support such conclusion.
(4) Cell viability assays: why two different cell lines (A549, MDA-MB-231) have been used to test the cytotoxic activity of RL2 dimer and of RL2 Ser mutant? It would be more conclusive to use the same cell line to compare the three different RL2 constructs.
(5) In the interpretation of cytotoxic activities, the authors might take into account the following points:
- To explain the similar cytotoxic activities of RL2 and dimeric RL2, the authors might consider that RL2 makes disulfide-bonded dimers in the cellular assays. Indeed, due to the pH and oxidative conditions of the cell incubation medium, it is likely that RL2 gets highly aggregated (as indicated by DLS measurements), an environment which should favour the formation of intermolecular disulfide bridges.
- A possible explanation to the loss of cytotoxic activity for RL2 Cys mutant could be that Cys thiol exchange plays a role in the internalisation mechanism, as shown by several groups: see Aubry et al (2009) FASEB J. 23, 2956-2967; Abegg et al. (2017) JACS 139, 231-238.
(6) In the analysis of CD data, the authors conclude that RL2 gets more folded upon heating, in an aqueous environment. This statement is somewhat counterintuitive. Have the authors tried to use deconvolution programs to quantify the proportion of secondary structures? The rise of the contribution around 220 nm could be ascribed to b-sheet aggregated species forming at high temperature. There could also be a contribution from polyproline helix structures due to the high number of proline residues in the sequence.
(7) The authors state several times in their manuscript that TFE is a membrane-mimetic environment. This is an over-simplistic assessment. The authors should use membrane mimetics such as lipid vesicles for CD measurements, or detergent micelles for solution NMR studies.
(8) One even more over-simplistic conclusion in page 12 (line 400) is the analysis of aggregation state “in early endosomes“. If this conclusion is only supported by in vitro DLS experiments at pH 5.5, the authors should take care not overinterpret their data.
Minor points:
(9) Abstract, line 22: I suggest to use the term “disulfide-bonded homodimer” that is more precise than “homodimer”
(10) Since the numbering of RL2 fragment differs from that of full-length casein, it might be helpful to precise that residue Cys8 in RL2 recombinant fragment corresponds to position Cys30 of human kappa-casein polypeptide (P07498 entry), in Fig 1 legend.
(11) The experimental section contains a few inaccuracies or omissions:
- Expression protocol, line 92: the authors indicate a concentration of 1 mM NaCl, is this correct ?
- Purification protocol, lines 94-96: the sentence should be rephrased as DEAE Sephadex is not an affinity chromatography but an ion exchange chromatography.
- MTSL labelling protocol: a buffer containing 4 M guanidinium chloride is used. The authors should precise at which stage of the protocol guanidinium chloride is removed.
- line 121 : “disodium salt of DSS“. This is not correct, it is a monosodium salt.
- lines 130-131, HNHA experiment: the measured coupling constant is not between 15Ni and HNi but between HNi and HAi spins.
(12) The unit on the y axis of CD spectra is not correct. It should be 10e4, not 10e-4.
(13) The authors should precise that the HSQC spectra are 1H-15N HSQC in the text and in the legend of figures.
(14) Table 2, supporting information: J coupling constants are provided with non significant digits. A single digit after the decimal point should be kept and the Hz unit should be mentioned.
(15) Supporting information: the word “fragment” is missing in the title.
(16) Figure S4 legend, replace “combination” by “superimposition”
typo highlited -> highlighted
Author Response

(The authors gave the same response as above.)

Round 2
Reviewer 3 Report
In this revised version of their manuscript, the authors have addressed most of my concerns.
However, the major point below is neither commented nor addressed by the authors:
(3) In the NMR study page 10, the authors state that “intense signals from the dimers in the spectra [are] absent”. This statement is not clear. In which protein preparation do the authors expect to have a significant dimer population?
I recommend to record an 1H-15N HSQC spectrum of the purified dimeric form to reveal any differences with the monomeric form.
Furthermore, it is written in conclusion, page 12 (line 405), that “NMR analysis of RL2 samples confirmed the presence of S-S homodimers”. I could not find any data in the manuscript to support such conclusion.
Minor corrections required for other addressed points:
(1) Page 3, line 110: precise the concentration of Tris buffer used
(13) The authors have added 1H-15N HSQC in the text. However in the legend of Figure 6, they have indicated 13C-15N instead of 1H-15N.
(15) The word “fragment” was missing in the title of Supplementary Information (page S1), not in Fig. S7 legend.
Author Response
We are thankful to reviewer for this comment. We are sorry that we forgot to reply to it in our previous verson of corrected paper.
Comment:
(3) In the NMR study page 10, the authors state that “intense signals from the dimers in the spectra [are] absent”. This statement is not clear. In which protein preparation do the authors expect to have a significant dimer population?
I recommend to record an 1H-15N HSQC spectrum of the purified dimeric form to reveal any differences with the monomeric form.
Reply to comment:
We agree with reviewer that it could be nice to record an 1H-15N HSQC spectrum of the purified dimeric form. Unfortunately, we could not perform this experiment, because now we have no 15N labeled RL2 and it will need quite some time and money to buy 15N media and to synthesize new sample in quantity which will be enough for separation of dimers.
Moreover based on the results of our experiments we are sure that if we have such sample we could not see any NMR spectra, because of following reason:
At pH 3.5 NMR spectra of RL2 sample (which consists of monomers, dimers and aggregates) fully coincide with NMR spectra of purified diamagnetic MTSL-labeled monomer RL2. This means that either NMR spectra of monomer and dimer are similar, or contribution of NMR spectrum of dimers is absent. At the same time DLS showed that aggregates and dimers are present at pH 3.5 and then they should contribute into NMR spectrum if only their relaxation times is not too short. But they do not contribute!
We explained it in paper as following:
“Taking into account the presence of oligomeric RL2 even under acidic conditions (Table 1) and the absence of non-covalent RL2-MTSL oligomers under the same conditions it is reasonable to assume that (i) either NMR spectra of monomers and dimers are similar or (ii) the signals from the dimers in NMR spectra are absent because of its greater tendency to aggregate compared to monomer even under acidic conditions (confirmed by DLS). As the relaxation times of monomers and dimers are different due to different sizes, it should be seen in NMR spectra which not the case. Thus from our point of view the second proposal is much more reasonable.”
Comment:
Furthermore, it is written in conclusion, page 12 (line 405), that “NMR analysis of RL2 samples confirmed the presence of S-S homodimers”. I could not find any data in the manuscript to support such conclusion.
Reply to comment:
We agree with review and change the sentence:
“NMR analysis of RL2 samples confirmed the presence of disulfide-bonded homodimer and RL2-BME adducts.”
to
“NMR analysis of RL2 samples confirmed the presence of RL2-BME adducts.”
Comment:
Minor corrections required for other addressed points:
(1) Page 3, line 110: precise the concentration of Tris buffer used
Reply to comment:
It is corrected.
“All buffers contained 0.1 M Tris-НСl pH 7.5, 2 M urea, 10 mM β-mercaptoethanol.”
Comment:
(13) The authors have added 1H-15N HSQC in the text. However in the legend of Figure 6, they have indicated 13C-15N instead of 1H-15N.
Reply to comment:
It is corrected.